# Awareness and Attitude of Polish Gynecologists and General Practitioners towards Human Papillomavirus Vaccinations

**DOI:** 10.3390/healthcare11081076

**Published:** 2023-04-10

**Authors:** Piotr Sypień, Weronika Marek, Tadeusz M. Zielonka

**Affiliations:** 1Sebastian Petrycy Health Care Facility, 33-200 DąbrowaTarnowska, Poland; 2Department of Obstetrics and Gynecology, Jagiellonian University Medical College, 31-501 Cracow, Poland; 3Department of Family Medicine, Medical University of Warsaw, 02-005 Warsaw, Poland

**Keywords:** papillomavirus infections, HPV vaccines, knowledge, prevention, control of cervical cancer, gynecologists, general practitioners

## Abstract

Human papillomavirus (HPV) leads to diseases of the skin and mucous membranes of the anogenital and upper gastrointestinal tract, especially neoplasia. HPV vaccinations effectively protect against the development of HPV-related diseases. However, despite the wide availability of vaccination for patients, only a few percent of Polish children are vaccinated. The reasons for this are certainly complex. Therefore, the aim of the study was to evaluate gynecologists’ and general practitioners’ knowledge, awareness and attitudes towards HPV vaccination and analyze their opinions about the interest in HPV vaccinations among children and parents. An anonymous, voluntary, cross-sectional survey study was conducted among 300 Polish gynecologists and general practitioners. Participants were from a diverse group with a wide range of work experience and different workplaces. Most of the respondents (83%), especially the gynecologists (*p* = 0.03), declared informing and discussing with parents the HPV-related diseases and prevention methods. Only 8% of the participants reported a negative reaction from parents when talking about HPV vaccines. However, in practice, physicians very rarely recommend this vaccine. HPV vaccination was recommended more often by female physicians (*p* = 0.03), general practitioners (*p* < 0.001), physicians working over five years (*p* < 0.001), doctors who regularly vaccinated themselves against influenza (*p* = 0.01) and those who vaccinated their children against HPV (*p* =0.001). The availability of educational materials for parents and/or patients encouraged physicians to provide this information (*p* < 0.001). Polish gynecologists and general practitioners declared a positive attitude regarding HPV vaccines; however, they rarely recommended this vaccine. Physicians who vaccinate themselves against influenza and their own children against HPV are more likely to provide information and encourage HPV vaccination in others. Additionally, the availability of educational material for parents and adolescents plays an essential role in the popularization of this vaccination. Knowledge alone is not enough for physicians to recommend the vaccination to patients.

## 1. Introduction

Human papillomavirus (HPV) infection is perceived as the most common sexually transmitted infection and it plays a crucial role in the formation of various neoplasm in women and men [1,2]. Diagnosis of HPV-related diseases is based primarily on the recognition of clinical signs of infection and the performance of HPV DNA testing in a genetic test. In addition, regular cytology (Pap smear) in women allows for the early detection of premalignant changes in the cervix (carcinoma in situ) [3]. However, these diagnostic methods have some limitations and do not eliminate the virus from the body.

The introduction of HPV vaccines has allowed for a successful decrease in the rate of HPV infections and HPV-related diseases [4]. Although the HPV vaccine has already been included in national immunization programs in more than 100 countries, significant differences are observed in vaccination coverage [5]. Previous studies show a high HPV vaccination coverage in most of the European Union countries. Between 2010 and 2017, HPV-vaccination coverage was more than 80% in Spain, Belgium, Finland, Sweden, Portugal, Hungary and Malta, and simultaneously between 51–70% in Denmark, Italy, the Netherlands, Ireland, the Czech Republic and Luxembourg [6]. Different mentalities are well illustrated by the example of Germany, where the percentage of vaccinated girls in the eastern federal states is much higher than in the western [7].

According to the United Nations International Children’s Emergency Fund (UNICEF) data, in Poland, only 22,342 boys and girls were vaccinated against HPV in 2018 (about 5% of the target population) [8]. At the same time, the current low HPV vaccination coverage deviates from other non-mandatory vaccines: 232,435 against *Streptococcus pneumoniae* (56%) and 105,603 against Rotavirus—around 25% of the targeted children’s population [8,9]. The reasons for such a low implementation are varied and not clearly defined. Some sparse previous studies of the Polish population have shown that firstly, a barrier in the popularization of HPV vaccination is the high cost of the vaccine and secondly, there is low public awareness about HPV-related diseases [10]. Currently, HPV vaccination in Poland is fully paid for by parents. Some local governments are organizing HPV vaccination campaigns covering the cost of the vaccine, but access to reimbursable vaccination is very limited. Additionally, the global anti-vaccination trend is quickly spreading in Poland despite confirmation of the safety and effectiveness of vaccines [11,12]. Also, the negative aspects of youth’s sexuality and sex education are particularly highlighted in some public discussions about HPV vaccines.

The annual incidences of cervical cancer are 19.8 cases per 100,000 women in Poland, 8.43/100,000 in Austria, 8.99/100,000 in the Netherlands, 10/100,000 in France and 7.62/100,000 in Cyprus [13]. This makes vaccination in Poland all the more important, as HPV-related diseases are more common than in other European Union countries.

One of the most important factors in popularizing HPV vaccination is the awareness and commitment of physicians. Physicians speaking with parents play an important role in the decision to vaccinate against HPV [14]. Victory et al. showed that more than 60% of parents who had received recommendations from a physician had vaccinated their children against HPV; however, out of those who had not received HPV vaccine recommendations from a physician, only 8% reported having vaccinated their children [15]. Pediatricians are mainly responsible for fulfilling the vaccination schedule for children in Poland [16]. However, the non-mandatory HPV vaccination requires a recommendation by doctors in other specialties, especially those who work closely with HPV-related diseases. Among them, gynecologists (GYNs) and general practitioners (GPs) are particularly important and can play an essential role in reducing the incidence of HPV-related diseases by recommending this vaccine [17]. Gynecologists mostly deal with the diagnosis and treatment of HPV-dependent diseases in Poland. Their task is to make young women aware of the need for diagnosis, treatment and prevention of HPV-related cancers. They play an important role in building gynecological social awareness. This may prompt women to vaccinate their children in the future, if discussed during visits to the GYNs. On the other hand, GPs deal with other problems than the gynecological manifestation of HPV-related infections, such as genital warts. In addition, just as pediatricians carry out the immunization program for children in Poland, their recommendation influences the parent’s decision to vaccinate.

The aim of the study was to evaluate GYNs’ and GPs’ knowledge, awareness and attitudes towards HPV vaccination and analyze their opinions about the interest in the HPV vaccinations among children and parents.

## 2. Materials and Methods

This cross-sectional survey study was performed between February 2018 and December 2018. The research participants included 300 Polish GYNs and GPs working inhospitals and outpatient clinics in Warsaw, Tarnow, and physicians from different cities in Poland participating in national medical conferences in Warsaw who voluntarily agreed to complete an anonymous survey (Table 1). The protocol of the study was approved by the Ethics Committee at the Medical University of Warsaw, Poland. The inclusion criteria were working as a GYN or GP and voluntary consent. We excluded from the study doctors who did not work daily with children, adolescents or young adults.

To achieve the research goals, the authors prepared a special questionnaire based on the literature review [10,18,19,20]. The questionnaire consisted of 29 questions with the possibility to choose one or more preferred answers. Questions were divided into three groups. The first six questions referred to socio-demographic and professional experience data. The second part of the survey comprised 18 questions about the doctor’s attitude and personal experience of the HPV vaccine, sources of knowledge, providing information about the HPV vaccine in their daily practice, their evaluation of parents’ interest and attitude towards the HPV vaccine and access to educational materials for them. It also asked about the HPV vaccination recommendation frequency in daily practice, reasons for recommending or not recommending vaccination, opinions about the funding status of the HPV vaccine, attitude towards the HPV vaccination of their own child and yearly influenza self-vaccination. The third part of the survey checked the doctor’s knowledge about the HPV vaccinations and consisted of five questions. The questions pertained to infection by HPV, efficacy of the HPV vaccine, the vaccination program and recommendation for vaccination according to gender and sexual activity.

Statistical analysis was performed with the use of Statsoft Statistica^®^ and Microsoft Excel^®^. Categorical data were presented as frequencies and percentages. In the study, we developed relationships in relation to awareness, knowledge, opinions about HPV of employees according to gender, place of work, and looked for other sources to account for the occurrence of certain views. To compare two groups of categorical variables, we used the chi-square test and we considered *p* values < 0.05 as statistically significant.

## 3. Results

### 3.1. Providing Information and Interest about Vaccination

Around 83% of respondents reported providing information about the HPV-related diseases and vaccines to young patients and their parents. However, the positive response rate varied depending on specialization, professional experience and place of work (Table 2). Physicians with more work experience regularly informed patients about the risks of HPV infection (*p* < 0.0001). GPs more often provided this information on their own initiative than GYNs did (Table 3). In turn, GYNs more often than GPs provided this information when asked by patients. Educational materials are more often available in GP practices (54% vs. 27%; *p* < 0.001) and have a positive effect on the frequency of providing information about vaccinations by physicians (95% vs. 74%; *p* < 0.001). Most participants rate the level of parents and adolescents’ interest in HPV vaccination as poor (Table 4).

### 3.2. HPV Vaccination Recommendations

Over 70% of respondents reported recommending HPV vaccines to their patients. This is especially true for women physicians, GPs and workers in outpatient clinics witha longer tenure (Table 5). The frequency of recommendations varied from a few times per year (41% of GYNs, 48% of GPs) to less than once per year (51% and 33%, respectively) to only a few times monthly (8%; 19%). Participants indicated various reasons for HPV vaccine recommendations (Table 6). Additionally, local government initiatives had a greater influence on GPs in comparison to GYNs (24% vs. 3%; *p* < 0.0001). Less than one third of participants (27%) did not recommend HPV vaccines in their practice. As the reason for not recommending the HPV vaccine, the doctors indicated: lack of confirmation of vaccine effectiveness (16%), side effects (10%), high cost (10%) and being against vaccinations in general (1%). One in four respondents gave additional answers regarding not working with relevant groups of patients to recommend this vaccination.

### 3.3. Doctors’ Attitude

Most respondents (75%) showed a positive attitude to vaccinate their own children, but the opinion on vaccination varied when considering the sex of the child (Table 7). Doctors with a positive attitude towards the vaccination of their own children more frequently informed parents about HPV vaccinations (87% vs. 67%; *p* = 0.0001) and recommended this procedure (82% vs. 50%; *p* < 0.0001). Almost half of the responders (47%) are vaccinated against seasonal influenza every year. These doctors more frequently spoke to patients about HPV vaccines and recommended them (81% vs. 68%; *p* = 0.01). Both GYNs and GPs are in favor of financial support for HPV vaccines (respectively, 65% and 83%). According to 28% of all participants, HPV vaccines should be compulsory and free of charge, for 47% the vaccine is recommended and co-financed by the Government, while for 25% of the population, the vaccine is recommended but not refunded. Compulsory status was more eagerly proposed by GPs than GYNs (75% vs. 19%, *p* = 0.003), respondents who were vaccinated against the flu (34% vs. 20%, *p* = 0.009) and those who work in a village or small town versus in a big city (36% vs. 22%, *p* = 0.04). Financial support was especially highlighted by GPs rather than the GYNs (81% vs. 64%, *p* = 0.006).

### 3.4. Knowledge about Vaccination

Only 42% of GYNs and 57% of GPs declared their cognizance concerning HPV vaccination as sufficient. Doctors indicated the most common sources of obtainingknowledge concerning HPV vaccination were the scientific materials and press (74% of GYNs, and 83% of GPs), medical conferences (70%; 68%, respectively), the internet (38%; 40%), pharmaceutical companies (42%; 21%), other doctors (24%; 17%), university education (28%; 18%), public media (4%; 7%) and product data sheets or clinical trials (2%; 1%).Most ofthe respondents were familiar with the high-risk groups of HPV infection, vaccination dose data and vaccination target, but they differed with the knowledge about vaccination protection (Table 8). Knowledge about all the potential vaccinations was significantly higher among physicians who provide information about vaccines in comparison to those who do not (94% vs. 74; *p* = 0.0008), possessing educational materials in the work place in comparison to those who do not have any (51% vs. 37%; *p* = 0.03), concerned with financial support for vaccines or not (82% vs. 69%; *p* = 0.02%) and mandatory status or not (36% vs. 23%; *p* = 0.02) and having a positive attitude towards vaccinating their own children in comparison with those with a negative attitude (87% vs. 69%, *p* = 0.001).

## 4. Discussion

The reasons for the differences in adolescent HPV vaccination coverage are complex and include issues of government policy, the organization of the health care system and physicians and parents’ attitudes [14,21]. The presence of national vaccination programs in most European countries increases the HPV vaccination coverage of the population [5]. In a successful immunization program, therefore, both sound government regulation and direct patient communication efforts are important.

The Polish long-term National Oncological Strategy plans to vaccinate at least 60% of youth against HPV by 2028 [22]. This is a challenge as the current vaccination coverage is around 8%. At the same time, the only change so far is that the recommended, but optional, anti-HPV vaccinations will only receive a 50%reimbursement of costs. Previous experiences with non-mandatory, recommended, but only partially reimbursed vaccines have not been successful in Poland. It has not been possible to achieve a satisfactory effect so far and further efforts are being performed to increase HPV-vaccination coverage. Information campaigns and free vaccination campaigns organized by local governments are receiving positive responses from the public, as emphasized especially by Polish parents [23]. Therefore, additional nationwide efforts are needed to transfer the success of the vaccination program to the entire population. Raising awareness and changing attitudes will help reduce the incidence of HPV-dependent diseases. Moreover, the COVID-19 pandemic has shown the significant reluctance of the Polish society to vaccinate even when their life is in danger [24]. The study showed that if we want to increase vaccination coverage in the whole society, we must start by increasing the coverage of doctors and their children. Therefore, special efforts should be made to encourage physicians to promote HPV vaccination in their daily practice. This study confirmed a huge discrepancy between the declared imparting of information about the importance of HPV infection and a very low uptake of HPV vaccination. A very important role in the promotion of immunization is played by the personal convictions and approach of physicians, which in our study was quantified by a positive attitude towards vaccinating their own children against HPV and regularly vaccinating themselves against influenza. These factors significantly increase their willingness to promote the HPV vaccination. Knowledge alone is not enough, and doctors must first be convinced of the vaccinations themselves and believe in their positive effects. The role of physicians in promoting vaccination is crucial. That is why undergraduate and postgraduate education in this field is so important. The study showed some shortcomings in the knowledge of doctors in the field of HPV-related diseases and appropriate prevention. Doctors can not only know, but also have to follow the recommendations themselves. Meanwhile, only a dozen or so percent of medical students in Poland and 20% of doctors regularly vaccinate themselves against the flu [25]. As our research has shown, this also has a negative impact on the HPV vaccination coverage of young people. Counteracting anti-vaccine movements should be started during medical school by creating appropriate attitudes among future doctors. Physicians who do not vaccinate themselves will not be persuasive when recommending vaccinations to their patients.

Access to professional and accurate information is a challenge for both patients and doctors. A particularly strong correlation was revealed between the accessibility of educational materials for parents in medical facilities and the frequency of recommendations of HPV vaccination by physicians. Brochures and leaflets are perceived as additional to a doctor’s recommendation and an effective method of education, and have a positive influence on Polish parents in making decisions about their child’s immunization [10,20].

The study demonstrated that Polish physicians are quite knowledgeable about HPV infections and vaccines, but some of them have shortcomings. However, it is necessary to emphasize that medical professionals who have to deal with vaccinations should be much more educated and aware of the problems than patients. Perhaps because of this, the willingness of doctors to recommend HPV vaccination is unsatisfactory. Interestingly, physicians in other countries also sometimes show passivity in recommending this vaccination against HPV [26,27]. In the study, GPs recommended HPV vaccination more often than GYNs. Additionally, the role of GPs in Poland in the popularization of medical knowledge is similar to those of specialists in other European countries [27]. Some limitations regarding low HPV vaccination coverage have already been indicated in other studies, such as the lack of time talking to patients about sex education and/or the anxiety felt when talking about sex [28,29,30]. Furthermore, other studies focusing on parental decisions indicated the cost of vaccination as a main obstacle in vaccinating their children [11,26]. Financing HPV vaccines has a positive cost-effectiveness for a country’s economy by increasing the number of individuals actively working and their quality-adjusted life expectancy [31,32]. However, price is not the only limitation of HPV vaccination. Our research shows, however, that for some doctors, the problem is a lack of appropriate knowledge, as they indicated ‘lack of vaccine effectiveness’ and its ‘side effects’ as the reason for not recommending these vaccinations. This confirms the need to accurately educate doctors in this field.

Although GYNs are not directly responsible for performing HPV vaccinations in Poland, this study showed that those specialists declared themselves as more eager to inform patients about the risks of HPV infection than general practitioners. They underscored the young patients’ interest in HPV vaccination more than GPs, which indicated their essential role in the introduction of this vaccination among youth. The low level of HPV vaccination among adolescents demonstrates the crucial role of parental education in this field. The leading role of GPs to promote this vaccination among parents was indicated in many previousstudies [1,33,34]. What is interesting is that female physicians have more awareness of HPV-related diseases. However, this does not translate to more frequent recommendation of HPV vaccination by women.

Another important aspect in promoting vaccination is years of service, which is associated with an increased willingness to recommend vaccinations. The rare recommendation of HPV vaccinations by young doctors indicates the shortcomings of university education, which are compensated for in postgraduate education and this needs to be changed. Physicians working in outpatient clinics have more opportunities to educate patients about disease prevention. General practitioners are more likely to talk to parents and know their views. Therefore, they are also more supportive of the option of mandatory HPV vaccination when inquiring what the vaccination status should be in Poland.

In order to increase the encouragement of vaccination by doctors, it is necessary, first of all, to build the right personal conviction for vaccines among physicians. It is not so much the knowledge of the physicians that is important, but their personal beliefs [32]. That is why it is so important to monitor doctors who are vaccinating their children as prescribed. These positive attitudes should be instilled in physicians from the early years of their university education and during their professional training.

The study has some limitations. It was conducted among a small population of well-educated physicians working mainly in two Polish regions. There are over 7000 GYNs and over 11,000 GPs in Poland. Therefore, it is more of a pilot study, although based on a randomly selected group in two towns. In the case of Warsaw, the participants constituted a smaller proportion of the whole, but in Tarnow, it was larger. The results are based on the physicians’ declarations and, despite the anonymous nature of this study, may be overstated. Some physicians may be hesitant to admit that they are not acting according to their knowledge and recommendations. The answers proposed in the survey did not cover all the options, but it is a consequence of the choice of simple closed questions. Voluntary participation in the survey may also create falsely inflated results, as those who are unsure of their competence or with negative attitudes to HPV vaccination may have deliberately refused to participate. Therefore, it is necessary to further analyze the awareness and attitudes of physicians in order to raise the level of HPV vaccination in Poland. This is additionally important because the public is positively disposed to vaccination, while there is inadequate awareness of the need [23,35]. The public trusts doctors, so overcoming the problems outlined by professionals should be thoroughly researched and solved in the future.

## 5. Conclusions

GYNs and GPs present satisfactory awareness and attitudes towards HPV vaccinations. However, they rarely recommend this vaccination to parents and patients. Therefore, the right motivation of doctors is needed. Their own positive attitude towards vaccinations is particularly important. Physicians who vaccinate their own children against HPV and vaccinate themselves against influenza are more likely to provide information and encourage vaccination in others. Additionally, the availability of educational material among parents and adolescents plays an essential role in the popularization of this vaccination. Knowledge alone is not enough for physicians to recommend vaccinations to patients. Time and determination are needed, resulting from the doctors’ profound belief in the need for HPV-related disease prevention.

## Figures and Tables

**Table 1 healthcare-11-01076-t001:** Characteristics of Medical Practitioners.

93 Characteristics	Category	Participants (*n* = 300)	%	GYNs (*n* = 144)	%	GPs (*n* = 156)	%
Gender:	Female	193	64	71	49	122	78
	Male	107	36	73	51	34	22
Origin	rural area	39	13	17	12	22	14
(residents):	less than 100,000	88	29	33	23	55	35
	100,000–500,000	85	28	35	24	50	32
	over 500,000	83	28	57	40	26	36
Work location	rural area	35	11	1	1	34	22
(residents):	less than 100,000	61	20	22	15	39	25
	100,000–500,000	102	34	48	33	54	35
	over 500,000	93	31	68	47	25	16
Working experience:	less than 5 years	51	17	32	22	19	12
	5–20 years	102	34	52	36	50	32
	20–35 years	124	41	53	37	71	46
	over 35 years	22	7	6	4	16	10
Working	clinical hospital	50	17	48	33	2	1
place *:	district hospital	31	10	25	17	6	4
	city hospital	66	22	57	40	9	6
	outpatient clinics	156	52	8	6	148	95
	private services	58	19	45	31	13	8

*—Some participants worked both in hospital and outpatient clinics.

**Table 2 healthcare-11-01076-t002:** Factors affecting the information provided by the doctor about HPV-related diseases and their prevention.

Characteristics	Category	Positive Answers	*p*
Specialization:	GYNs	136 (87%)	0.03
	GPs	111(77%)	
Work experience:	less than 5 years	29 (57%)	0.0001
	more than 5 years	219 (88%)	
Work place:	hospital	110 (75%)	0.04
	outpatient clinics	133 (85%)	

**Table 3 healthcare-11-01076-t003:** Reasons for providing HPV information.

Statement	Number and Percentage of Positive Answers		
	All Participants	GYNs	GPs
Doctor’s own initiative	151 (50%)	52 (33%)	99 (69%)
Parental interest	82 (27%)	17 (11%)	65 (45%)
Adolescent/Patient interest	104 (35%)	94 (60%)	10 (7%)

**Table 4 healthcare-11-01076-t004:** Doctor’s opinion on the interest of parents and their children in HPV vaccinations.

Level of Interest	Group	Statement		
		All Participants	GYNs	GPs
none	parents	55 (18%)	41 (28%)	14 (10%)
	adolescents	153 (17%)	40 (26%)	13 (36%)
small	parents	143 (48%)	67 (43%)	76 (53%)
	adolescents	154 (51%)	82 (53%)	72 (50%)
medium	parents	89 (30%)	40 (26%)	49 (34%)
	adolescents	48 (16%)	31 (20%)	17 (12%)
high	parents	9 (3%)	5 (3%)	4 (3%)
	adolescents	4 (1%)	2 (1%)	2 (2%)

**Table 5 healthcare-11-01076-t005:** Physicians’ willingness to recommend HPV vaccination.

Parameter		Number and Percentage	*p*
Gender	female	150 (78%)	0.03
	male	71 (66%)	
Specialization	GP	112 (83%)	0.001
	GYN	98 (63%)	
Years of service	<5 years	22 (43%)	0.001
	>5 years	200 (80%)	
Work place	hospital	88 (60%)	0.01
	outpatient clinic	122 (78%)	

**Table 6 healthcare-11-01076-t006:** Reasons for HPV vaccine recommendations by physicians.

Parameter	Number	%
own knowledge and belief	186	62
experts’ recommendations	132	44
pharmaceutical companies	57	19
local government initiatives	42	14
colleagues’ opinion	24	8
supervisors’ advice	9	3

**Table 7 healthcare-11-01076-t007:** Doctors’ individual approach towards HPV implementation for their own child.

Statement	Number of Positive Answers	%
I did not vaccinate or I will not vaccinate	19	7
I vaccinated my daughter(s)	56	19
I vaccinated my son(s)	15	5
I will vaccinate my daughter(s)	108	36
I will vaccinate my son(s)	58	19
I would vaccinate my daughter(s) but this vaccine was unavailable on the Polish Market	34	11
I would vaccinate my son(s) but this vaccine was unavailable on the Polish Market	10	3

**Table 8 healthcare-11-01076-t008:** Distribution of correct answers among examined physicians.

Question	All	GYNs	GPs	*p*
Who can be infected by HPV?	201 (67%)	98 (68%)	103 (66%)	0.71
Against which disease(s) does the HPV vaccine protect?	86 (29%)	43 (30%)	43 (28%)	0.73
How many doses concludes a full HPV vaccination schedule?	262 (87%)	125 (87%)	137 (88%)	0.79
Who should be vaccinated?	215 (72%)	110 (78%)	105 (67%)	0.08
When should people be vaccinated?	236 (79%)	102 (71%)	134 (86%)	0.001

## Data Availability

The research data are in the possession of the authors and can be shared upon request.

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
