# Peer review of "Awareness and Attitude of Polish Gynecologists and General Practitioners towards Human Papillomavirus Vaccinations"

_healthcare, 2023, doi:10.3390/healthcare11081076_

Round 1
Reviewer 1 Report
Congratulations to the authors for their concise description about HPV vaccination status and implications in Poland. The manuscript is overall very well written. As HPV vaccination plays an important role in the prevention of several cancers, I can only support studies that aim to increase vaccination rates.
I have a few comments for the authors to be addressed at this time.
- In the introduction the authors mention that gynaecologists play an important role in the recommendation of HPV vaccines. Although, if a woman goes first to the gynaecologists, HPV vaccine is often already too late. So please specify why gynaecologists play an important role (recommendation of vaccines through parents, explication of possible HPV-related cancers ..)
- In the methods you state that physicians voluntary agreed to fulfill the questionnaire, so in my opinion there is a positive inclusion BIAS of participants who were willing to complete the survey and are maybe more interested in HPV vaccines. Mention this in the discussion limitations part.
- In table 8 the authors pose the question “against which disease does HPV vaccine protect”? Please mention also penile cancer and head and neck squamous cell carcinoma. The latter is the most frequent HPV-related disease with a high morbidity.
- In the discussion the letter size suddenly changes.
- In the discussion on page 8 the authors mention that: “It’s not about the knowledge of doctors that is important but more about their personal beliefs.” I think you should delete this sentence since earlier, on the same page the authors mention that there is a lack of knowledge about vaccine effectiveness and side effects for not recommending the vaccine. These statements contradict each other.
- It should be useful to spend a little part in the discussion about changes or implementations that could be beneficial to increase the rate of HPV vaccination in Poland.
Author Response
Thank you very much for your review. We are trying to raise the HPV vaccination coverage, which is low and still needs a lot of work to be done to get closer to the average for European Union countries.
In the introduction the authors mention that gynaecologists play an important role in the recommendation of HPV vaccines. Although, if a woman goes first to the gynaecologists, HPV vaccine is often already too late. So please specify why gynaecologists play an important role (recommendation of vaccines through parents, explication of possible HPV-related cancers)
We agree with this opinion and have added it to the text as requested. They constituted one of the three surveyed groups, as we also assess the awareness of general practitioners and paediatricians.
In the methods you state that physicians voluntary agreed to fulfill the questionnaire, so in my opinion there is a positive inclusion BIAS of participants who were willing to complete the survey and are may be more interested in HPV vaccines. Mention this in the discussion limitations part.
You are right. We added it in the discussion as requested.
In table 8 the authors pose the question “against which disease does HPV vaccine protect”? Please mention also penile cancer and head and neck squamous cell carcinoma. The latter is the most frequent HPV-related disease with a high morbidity.
We revised table 8. Penile cancer and squamous cell carcinoma of the head and neck were included in the question.
In the discussion the letter size suddenly changes.
You're right, we changed it. The text has been one again edited.
In the discussion on page 8 the authors mention that: “It’s not about the knowledge of doctors that is important but more about their personal beliefs.” I think you should delete this sentence since earlier, on the same page the authors mention that there is a lack of knowledge about vaccine effectiveness and side effects for not recommending the vaccine. These statements contradict each other.
Yes, we corrected it. Not only knowledge but also beliefs.
It should be useful to spend a little part in the discussion about changes or implementations that could be beneficial to increase the rate of HPV vaccination in Poland.
We added as requested.
Reviewer 2 Report
This was an interesting study looking at the knowledge and attitudes of Polish doctors towards the HPV vaccine. It was a good study and the findings were interesting since the practices towards HPV vaccination in Poland did not reflect the findings of the study. Perhaps the authors can dig deeper to explore the reasons for that in future studies. This could be included in recommendations in the end.
Author Response
Comments to reviewer 2.
Thank you for your comment. We hope that the exact knowledge of the causes of low HPV vaccination will allow us to achieve a good result in the future. We have completed the data as requested.
Reviewer 3 Report
The study had a small interest for publication in a scientific journal. There are already dozens of articles on the same subject.
The authors should send their manuscript to a national journal. Moreover, the methodology is very deficient in its description, statistical analysis only descriptive is of little interest.
The manuscript does not respect the instructions to the authors of the journal. Authors should rewrite their articles, detail their methodology in a much more complete way and complete their scientific analysis.
Author Response
Thank you for your substantive review. It is difficult to change the study design at the review stage, but we tried to clarify it. However, we would like to point out that the articles part of a broader scientific study, the other elements of which have been integrated into the work and are indicated by a citation. Despite a number of attempts, it has still not been possible to convince the public about HPV vaccination. Therefore, in our project we under took to analyze awareness, attitudes and knowledge towards HPV vaccination in Poland. That's why we investigated doctors (pediatricians, general practitioners and gynecologists), adolescents and parents. Surveys of doctors are an important input and the results on the opinion of health providers in Poland and Central and Eastern Europe are limited. This is especially true for gynecologists, who in Poland are involved in screening, diagnostics and treatment of female HPV-related cancers.
We corrected the manuscript due to instructions for authors and we modified the methodology. The text was checked by English native speaker.
The level of HPV vaccinations in Poland is much lower than in other European Union countries, while the incidence of HPV-related cancers is higher. This poses serious social and medical problems. Importantly, the vaccination trend is not changing from many years, and there is growing concern about the willingness to refuse vaccinations. Therefore, we are very keen to present the problem as widely as possible. We were able to find new factors playing a role in vaccination coverage that were not previously mentioned. We show that education and free vaccination are not enough.
Round 2
Reviewer 3 Report
thanks to the authors for their additions and clarifications.
the article seems to me publishable in this form